# Enhanced nonlinear optomechanics in a coupled-mode photonic crystal device

Roel Burgwal [1,2] & Ewold Verhagen [1,2] ✉

The nonlinear component of the optomechanical interaction between light and mechanical vibration promises many exciting classical and quantum mechanical applications, but is generally weak. Here we demonstrate enhancement of nonlinear optomechanical measurement of mechanical motion by using pairs of coupled optical and mechanical modes in a photonic crystal device. In the same device we show linear optomechanical measurement with a strongly reduced input power and reveal how both enhancements are related. Our design exploits anisotropic mechanical elasticity to create strong coupling between mechanical modes while not changing optical properties. Additional thermo-optic tuning of the optical modes is performed with an auxiliary laser and a thermally-optimised device design. We envision broad use of this enhancement scheme in multimode phonon lasing, two-phonon heralding and eventually nonlinear quantum optomechanics.

The field of cavity optomechanics studies the interaction between a light field and mechanical vibration. On the one hand, the optomechanical interaction imprints the mechanical motion onto the light field, enabling extremely precise optical detection of position and spurring the development of highly precise sensors that approach and even evade fundamental measurement limits set by quantum mechanics[1–3]. At the same time, the light field can be used to manipulate the state of the mechanical resonator, which has allowed the creation of mechanical quantum states for use in quantum information technology, as information storage or as tool in the conversion of superconducting microwave qubits to optical qubits[4–8].

Many especially exciting applications have been envisioned that exploit nonlinear interaction between the light field and mechanical modes. The optomechanical interaction is in fact inherently nonlinear, but for current systems the linear component is dominating for quantum-level mechanical motion. A sufficiently strong nonlinearity would open up possibilities such as measurement-based non-classical state generation[9], energy-squeezed states[10], quantum non-demolition (QND) measurement of phonon number[11,12] or the photon-blockade effect[13,14]. Such effects become apparent in the single-photon strong coupling (SPSC) limit $g_0/\kappa > 1$, where $g_0$ is the optomechanical vacuum coupling rate, and $\kappa$ is the decay rate of the optical resonator. Although

this limit has been reached in atom-optomechanical systems[15], these systems do not satisfy the important condition of sideband resolution. Creating systems that meet both requirements remains a worthwhile pursuit.

Nonlinear optomechanical effects can be enhanced in a system of two coupled optical modes, both optomechanically coupled to one mechanical mode, often referred to as the membrane-in-the-middle (MIM) system[11]. This enhancement is particularly interesting for systems that approach the SPSC regime, as it makes nonlinear quantum effects more pronounced[14,16]. However, enhancement of the nonlinearity in such a multimode system over its magnitude in a comparable single-mode system is only possible when fulfilling two requirements on the system parameters, namely that the coupling rate between the two optical modes $J_O$ has to equal half the mechanical frequency $\Omega$ ($J_O = \Omega/2$), and that the mechanical frequency is larger than the optical decay rate $\kappa$ ($\Omega > \kappa$), i.e. the sideband resolution condition[17].

There have been many realisations of the MIM and related systems, in membranes[11,18], microtoroids[19,20], photonic crystals[21–23], ultra-cold atoms[24] and levitated particles[25]. However, the systems in which nonlinear transduction was studied did not have the required inter-mode optical coupling ($J_O \approx \Omega/2$) and optical decay rate ($\Omega > \kappa$) to

---

[1]Department of Applied Physics and Eindhoven Hendrik Casimir Institute, Eindhoven University of Technology, P.O. Box 513, 5600 MB Eindhoven, The Netherlands. [2]Center for Nanophotonics, AMOLF, Science Park 104, 1098 XG Amsterdam, The Netherlands. ✉e-mail: verhagen@amolf.nl

exhibit nonlinear effects that were enhanced above the intrinsic optomechanical nonlinearity and a direct experimental comparison is lacking. Realising the required parameters in a photonic crystal system would be specifically interesting, as these are receiving strong attention due to their large optomechanical coupling, small footprint, and compatibility with cryogenic operation[7,8,26]. Importantly, such multi-mode photonic crystal systems would also be useful to enhance linear transduction per input optical power[27].

Here, we describe a coupled-mode optomechanical crystal device fulfilling all the above requirements, and use it to demonstrate for the first time enhanced nonlinear optomechanical coupling in a direct comparison to a single-mode configuration in the same device. To do so, we measure nonlinear transduction of thermomechanical motion in a coupled-mode device, where one of the optical modes can be selectively and actively detuned to switch between a single and coupled-mode configuration. In addition, we quantify the enhancement of linear transduction with respect to input power that is also present in these systems, thus demonstrating two main advantages of the coupled-mode system. Our device shows strong coupling of optical and mechanical modes, for which we explored the use of mechanical anisotropy of silicon to tune mechanical properties without affecting the optical properties of the device. Finally, post-fabrication tuning of the optical modes, needed to correct inevitable fabrication imperfections, is achieved using thermal tuning with a laser as heat source and a thermally-optimised device design.

## Results

### Model

In the MIM system, two optical modes with annihilation operators $\hat{a}_L$, $\hat{a}_R$ and frequency $\omega$ couple to each other with rate $J_O$, and optomechanically to a mechanical mode with unitless position operator $\hat{x} = \hat{b} + \hat{b}^\dagger$, $\hat{b}$ being the mechanical annihilation operator, with vacuum coupling rates $g_L$, $g_R$. This creates a Hamiltonian (setting $\hbar = 1$)[28]

$$\hat{H} = (\omega + g_L\hat{x})\hat{a}_L^\dagger\hat{a}_L + (\omega + g_R\hat{x})\hat{a}_R^\dagger\hat{a}_R - J_O(\hat{a}_L^\dagger\hat{a}_R + \hat{a}_R^\dagger\hat{a}_L) + \hat{H}_m, \quad (1)$$

where $\hat{H}_m = \Omega\hat{b}^\dagger\hat{b}$ is the mechanical Hamiltonian with $\Omega$ the mechanical frequency. By moving to a basis of odd and even optical supermodes $\hat{a}_{e(o)} = 1/\sqrt{2}(\hat{a}_L \pm \hat{a}_R)$, the Hamiltonian can be written as

$$\begin{aligned}\hat{H} = (\omega - J_O)\hat{a}_e^\dagger\hat{a}_e + (\omega + J_O)\hat{a}_o^\dagger\hat{a}_o + \hat{x}\frac{g_L + g_R}{2}(\hat{a}_e^\dagger\hat{a}_e + \hat{a}_o^\dagger\hat{a}_o) \\ + \hat{x}\frac{g_L - g_R}{2}(\hat{a}_e^\dagger\hat{a}_o + \hat{a}_o^\dagger\hat{a}_e) + \hat{H}_m,\end{aligned} \quad (2)$$

describing a new system with two optical eigenmodes separated in frequency by $2J_O$. We consider $g_L = g_R$ ($g_L = -g_R$), in which situation we call the mechanical mode described by $\hat{x}$ even (odd). For an even mode, we have optomechanical interaction terms of the form $\hat{x}\hat{a}_{e(o)}^\dagger\hat{a}_{e(o)}$, of similar form to the canonical, single-mode optomechanical system. However, for an odd mode, interaction terms are of the form $\hat{x}\hat{a}_{e(o)}^\dagger\hat{a}_{o(e)}$, so-called *cross-mode* interactions.

For an odd mechanical mode, under the condition of slow mechanical motion $\Omega \ll J_O$, it is possible to diagonalise the Hamiltonian to isolate the quadratic coupling

$$\hat{H}_{int} \approx \frac{(g_L + g_R)^2}{8J_O}\hat{x}^2(\hat{a}_o^\dagger\hat{a}_o - \hat{a}_e^\dagger\hat{a}_e), \quad (3)$$

which promises a large nonlinear interaction for small $J_O$[11]. However, it was found early on that this form fails to capture remaining linear interaction[29,30], which precludes many applications such as a measurement of phonon number without reaching the SPSC limit. Moreover, it was shown that, in order for nonlinear interaction to be enhanced, sideband resolution $\Omega > \kappa$ and a specific optical coupling rate of $J_O \approx \Omega/2$ is required[14,16,17].

To describe both linear and nonlinear transduction fully, we solve the Langevin equations of motion derived from the Hamiltonian in Eq. (1), with operators replaced by their expectation values, $a = \langle\hat{a}\rangle$. The equations are solved perturbatively to second order to capture nonlinear effects, working in a frame rotating with the optical input field frequency $\omega_{in}$. The perturbative approach assumes that the mechanical motion is small, i.e. $\sqrt{\langle\hat{x}^2\rangle} < \kappa/g_0$, which is true for thermal motion in most of current optomechanical devices.

Using these equations, expressions can be derived (see 'Methods' for details) for the photocurrent $I$ power spectral density (PSD) $S_{II}[\omega]$ of heterodyne detection of light reflected from the optomechanical cavity, which can be compared to spectrum analyser measurements described below. The mechanical mode is assumed to be odd ($g_R = -g_L = g$) and driven only by the thermal environment, while only the left optical mode is probed. Then, for linear transduction in a single-mode device, the heterodyne PSD can be approximated by

$$S_{II}^{lin}[\Omega] = \kappa_{ex,L}^2 g^2 n_{in} n_{th} \left|\frac{\chi(-\Omega)}{\Delta_L(-\Omega + \Delta_L)}\right|^2, \quad (4)$$

while for the coupled-mode system, it reads

$$S_{II}^{lin}[\Omega] = \frac{\kappa_{ex,L}^2 g^2 n_{in} n_{th}}{4} \left|\frac{\chi(-\Omega)}{(-\Omega + \Delta_L + J_O)(\Delta_L - J_O)}\right|^2, \quad (5)$$

where we have introduced the complex detuning $\Delta_{L(R)} = (\omega_{in} - \omega_{L(R)}) + i\kappa_{L(R)}/2$, which contains as a real part the left (right) laser-cavity detuning, and as imaginary part contains the optical decay rate $\kappa_{L(R)}$, and $\kappa_{ex,L(R)}$ is the outcoupling rate of the cavities to their respective read-out ports. For simplicity, we assume that all optical decay rates in both systems have equal value $\kappa$ and that for the coupled system $\Delta_L = \Delta_R = \Delta$. The average amount of thermal phonons $n_{th}$ can be expressed as $n_{th} \approx k_B T/(\hbar\Omega)$, with $k_B$ the Boltzmann constant, $T$ the temperature, and $\chi(\omega) = 2\sqrt{\Gamma}\Omega/(\Omega^2 - \omega^2 - i\omega\Gamma)$, the mechanical susceptibility, with $\Gamma$ the mechanical decay rate, ignoring here optomechanical backaction effects on the mechanical mode at high powers for simplicity. Finally, $n_{in}$ is the amount of photons per second in the optical input field. In the coupled system, for optimal $\text{Re}(\Delta) = J_O$ and $J_O = \Omega/2$, both terms in the denominator can be minimised simultaneously and transduction at the mechanical frequency $\Omega$ reaches

$$\max(S_{II}^{lin}[\omega]) = \frac{16\kappa_{ex,L}^2 g^2 n_{in} n_{th}}{\Gamma\kappa^4}. \quad (6)$$

Compared to optimal linear transduction in a single-cavity system, where it is not possible to minimize both terms in the denominator simultaneously, that gives an enhancement of optomechanical sideband power of

$$\mathcal{E}^{lin} = \frac{\max(S_{II}^{lin}[\Omega])_{coupled}}{\max(S_{II}^{lin}[\Omega])_{single}} = \left(\frac{\Omega}{\kappa}\right)^2. \quad (7)$$

Thus, for equal optical input power $\mathcal{P}_{in} = \hbar\omega_{in}n_{in}$, the coupled-mode system can improve linear optical read-out of mechanical motion. The creation of fluctuations in the cavity field through the optomechanical interaction can also be viewed as the inelastic scattering of light from the input frequency to sidebands at frequencies $\Omega$ lower or higher than $\omega_{in}$. In this picture, the linear enhancement in coupled-mode systems can be regarded as using the two optical supermodes to achieve simultaneous resonance of both the input field and the optomechanically scattered sideband[27]. As a result, the intracavity photon number is larger in the coupled cavity case than in the single cavity pumped at equal power $\mathcal{P}_{in}$.

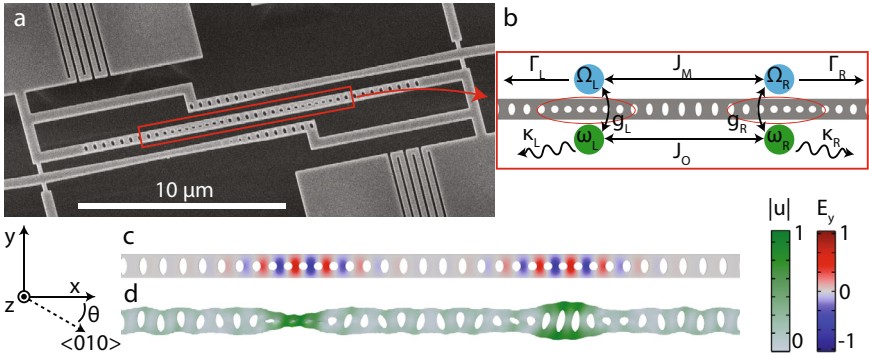

**Fig. 1 | Coupled-mode device design. a** A scanning electron microscope image of a fabricated device with nanobeam, waveguides and support structure. **b** Schematic of the mode frequencies and coupling rates in the double-cavity nanobeam. **c** The simulated $y$-component of the electrical field for the even optical supermode. **d** The

displacement ($|u|$) profile of a simulated mechanical supermode, which is an odd combination of the left and right cavity modes. Note that the depicted amplitude of motion is largely exaggerated. An angle of $\theta = 20°$ between the crystal axis and nanobeam was used for this simulation.

Nonlinear optomechanical interaction manifests itself as fluctuations in the reflected light at twice the mechanical frequency. Such fluctuations can be calculated when solving the EOMs to second order, where they give a detector PSD that can be simplified for a single-mode system to

$$S_{II}^{qua}[2\Omega] = \frac{\kappa_{ex,L} g^4 n_{cav} n_{th}^2}{\pi} \int_{-\infty}^{\infty} d\omega' \left| \frac{\chi(\omega')\chi(-2\Omega - \omega')}{(-2\Omega + \Delta_L)(-2\Omega - \omega' + \Delta_L)} \right|^2, \quad (8)$$

while for a coupled-mode system it reads

$$S_{II}^{qua}[2\Omega] = \frac{\kappa_{ex,L} g^4 n_{cav} n_{th}^2}{2\pi} \int_{-\infty}^{\infty} d\omega' \left| \frac{\chi(\omega')\chi(-2\Omega - \omega')}{(-2\Omega + \Delta + J_O)(-2\Omega - \omega' + \Delta - J_O)} \right|^2, \quad (9)$$

where $n_{cav}$ is the amount of photons in the cavity. For derivations and further details, see the 'Methods' section. In contrast to the linear enhancement, the nonlinear enhancement persists when normalising to the amount of photons in the cavity, which is the limiting factor in many experiments[7,26,31]. Again, the nonlinear transduction can be optimised in the coupled-mode system for the resonance condition $J_O = \Omega/2$, $\mathrm{Re}(\Delta) = J_O + \Omega$, where it reads

$$\max(S_{II}^{qua}[2\Omega]) \approx \frac{64 \kappa_{ex,L} g^4 n_{cav} n_{th}^2}{\Gamma \kappa^4}. \quad (10)$$

Comparing this to optimal transduction in a single-cavity system, for which only one term in the denominator can be minimised, we find an enhancement of nonlinear transduction given by

$$\mathcal{E}^{qua} = \frac{\max(S_{II}^{qua}[2\Omega])_{coupled}}{\max(S_{II}^{qua}[2\Omega])_{single}} = 2\left(\frac{\Omega}{\kappa}\right)^2. \quad (11)$$

This factor captures the optimal enhancement of nonlinear transduction possible in the coupled-mode system, which we find limited by the degree of sideband resolution $\Omega/\kappa$ of the system. As we will discuss in more detail later, the minimisation of both terms in the denominator of Eq. (9) can be understood as simultaneous resonance of the linearly scattered (the first sideband) and nonlinearly scattered light (the second sideband) with one of the optical supermodes.

### Coupled-mode design principle

As a basis for our coupled-mode device, we use a one-dimensional optomechanical crystal nanobeam in which an optical and mechanical mode are co-localised in a defect or cavity region to create a large optomechanical coupling[32]. Used often in recent quantum

optomechanics experiments[4,5,7,8,26,31,33,34], this cavity is particularly attractive because of its large optomechanical coupling $g_0$, operation in sideband-resolved regime $\Omega > \kappa$ and potential for ground-state initialisation in a cryogenic environment because of its high mechanical frequency ($\Omega/(2\pi) \approx 5$ GHz). Building a coupled-mode system from such favourable single-cavity building blocks ensures best possible performance of the coupled system. Starting from this basis, we create two optomechanical cavities by writing two crystal defect regions in the same nanobeam (see Fig. 1a and b). Through overlap of the evanescent fields of the cavity modes, couplings between the two optical as well as the two mechanical modes are created, characterised by inter-cavity coupling rates $J_O$ and $J_M$, respectively. The mode frequencies (decay rates) are given by $\omega_i$ ($\kappa_i$) and $\Omega_i$ ($\Gamma_i$) for the optical and mechanical modes, respectively, where $i \in \{L, R\}$ indicates the left and right cavities. Furthermore, we include next to our nanobeam two waveguides that allow us to couple to either the right or left cavity individually. These waveguides in turn connect to a dimpled, tapered optical fibre (see Fig. 1a)[35]. The cavity-waveguide coupling rates are given by $\kappa_{ex,i}$.

If the inter-cavity coupling exceeds the decay rates of the modes, as well as any possible frequency difference between the two modes, the local optical or mechanical modes hybridise into odd and even combinations of the left and right cavities, which are split in frequency by $2J_O$ or $2J_M$. Using finite-element method (FEM) simulations, we calculate the optical eigenmode frequencies of a nanobeam design and deduce $J_O$ from the supermode frequency difference. In Fig. 1c, an example of a simulated optical supermode is plotted. As we require $2J_O = \Omega$ for optimal enhancement of optomechanical effects, accurate control over the optical coupling rate is crucial. Coupling rates can be varied by changing the number and shape of the holes that make up the optomechanical crystal between the cavities, i.e. the coupling region.

After coupling region optimisation for optical coupling rate, the device design has a mechanical coupling rate that will typically not allow for strong coupling of the mechanical modes, as fabrication imperfections induce random frequency differences between the two mechanical modes that have to be overcome by a sufficiently large coupling rate. For independent tuning of the mechanical coupling rate, we exploit the anisotropy of the mechanical properties of the device material, monocrystalline silicon. By fabricating devices at an angle $\theta$ to the $\langle 010 \rangle$ crystal axis, the mechanical properties can be varied while leaving optical properties unaltered.

We studied the behaviour of mechanical modes as a function of fabrication angle $\theta$ using FEM simulations (see Supplementary Note 5 for details). For a non-zero angle, the $y$-symmetry of the system (orthogonal to beam axis, in plane) is broken due to anisotropy.

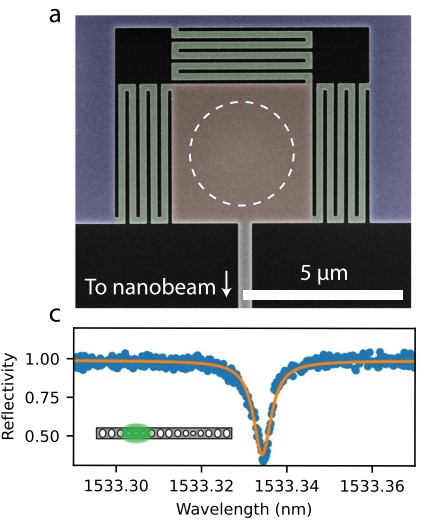

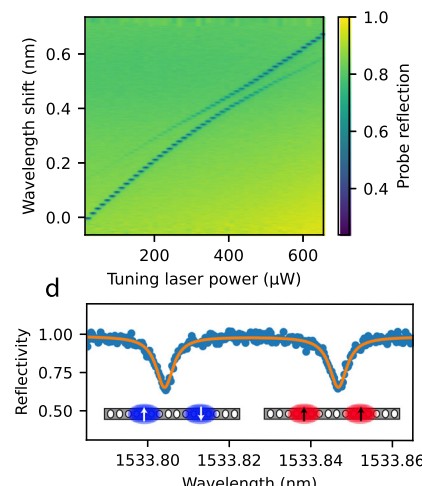

**Fig. 2 | Thermal tuning of the optical modes for frequency matching of coupled modes. a** False-coloured scanning electron microscope image of the thermally-optimised tuning structure connecting device (bottom) and substrate (top, purple). The meandering tethers (green) limit the flow into the substrate of heat generated by the green laser that is focused on the pad (orange, white circle indication of laser focus scale). **b** Spectrogram of IR probe reflection for varying power of the tuning laser, showing how a localised mode is tuned to form supermodes at the anticrossing with the other cavity optical mode. **c** An example optical reflectivity trace without thermal tuning. **d** Reflectivity trace with optical modes tuned to the same frequency. Reflectivity traces have been normalised to 1 for off-resonant optical input.

Although the nanobeam cavities are designed to localise the $y$-symmetric (*breathing*) mode, other modes exist at the same frequency with $y$-antisymmetry that are not confined by the defects. The introduction of non-zero $\theta$ mixes the breathing and $y$-antisymmetric modes and thus produces a new mode which can leak from the cavity more easily, resulting in a stronger effective $J_M$ and the formation of supermodes, of which an example is plotted in Fig. 1d. Note that the presence of higher-order single-cavity mechanical modes means that more than one pair of supermodes can be created. Altogether, our simulations indicate that the angle $\theta$ can be used to create a set of even and odd mechanical supermodes that will persist in the presence of fabrication imperfections.

## Optical strong coupling with active control

The devices are fabricated in 220 nm thick underetched silicon (see 'Methods' section for details). Due to fabrication imperfections, the actual optical resonance wavelengths of the left and right modes vary randomly with a typical difference of the order of 1 nm for a design wavelength of 1550 nm. As such a detuning will generally prevent the optical modes from hybridising and precludes enhancement effects, a post-fabrication tuning technique is needed. To allow active tuning, we exploit the temperature dependence of the material refractive index, which in turn controls the resonance wavelength. Creating a variable thermal gradient over the two cavities then allows for control of the inter-cavity detuning[36].

Here, we create a thermal gradient by illuminating the support structure at one end of the nanobeam with a 532 nm green laser spot. We design the support structure (see Fig. 2a) to optimise the strength of the achieved temperature gradient. Where the device connects to the support structure, a square pad is thermally isolated from the rest of the sample by thin, meandering tethers. These tethers limit the flow of laser-generated heat into the sample, allowing the suspended device to reach a higher temperature and thus significantly improving the tuning range. See Supplementary Note 4 and Supplementary Fig. 2 for thermal simulations of the support structure.

We characterise the device optical properties by a measurement of reflectivity through one of the waveguides coupled to a single cavity. For an untuned device, the reflectivity typically shows one, localised, optical mode (see Fig. 2c), here with $\kappa/(2\pi) = 632$ MHz and

$\kappa_{ex}/(2\pi) = 120$ MHz. When the tuning laser is applied, the resonant wavelength increases, at a faster rate for the mode closest to the chosen heating pad than for the distant mode. For the correct tuning, this will lead to an anticrossing between the left and right cavity modes (see Fig. 2b). At this point, two optical modes ($\kappa_o/(2\pi) = 642$ MHz, $\kappa_e/(2\pi) = 653$ MHz) are visible through our interrogation of a single cavity (see Fig. 2d), demonstrating the formation of two delocalised supermodes. From the minimal distance between the two supermodes, the inter-cavity coupling can be extracted to be $2J_O/(2\pi) = 5.4$ GHz, a value which is less than one optical linewidth away from the mechanical frequency at around 5 GHz. This, together with a large sideband resolution factor of $\Omega/\kappa \approx 7.5$, means that our device is capable of enhancing linear and nonlinear optomechanical transduction.

Importantly, we see no significant broadening of the optical modes with increasing tuning power, indicating that the tuning laser does not induce additional optical absorption and that fluctuations in the tuning are of a size well below the optical linewidth. Note that, with the optical modes tuned, the drop in reflectance on resonance is less pronounced than for a detuned device, because the effective out-coupling of a supermode to a single waveguide is lower than for a localised mode, moving the undercoupled device further away from critical coupling ($\kappa_{ex} = \kappa/2$). This could be overcome by changing the designed outcoupling rate of the device accordingly.

## Linear transduction of mechanical supermodes in a coupled-mode system

We now study the effect of multiple optical modes on the transduction of mechanical motion. We measure the thermomechanical motion at the mechanical frequency $\Omega$, which has an average amplitude that remains constant between different measurements (see Discussion section). In this way, it allows us to compare the strength of the optomechanical transduction of mechanical motion between detuned and tuned systems.

We use a setup that directly detects intensity fluctuations in the reflected light, using an erbium-doped fibre amplifier (EDFA) and a fast photodiode (see 'Methods' section for details). In Fig. 3a, we plot the photocurrent power spectral density (PSD) for the system with detuned optical modes while varying the detuning between the infrared laser and optical mode. We observe three mechanical modes,

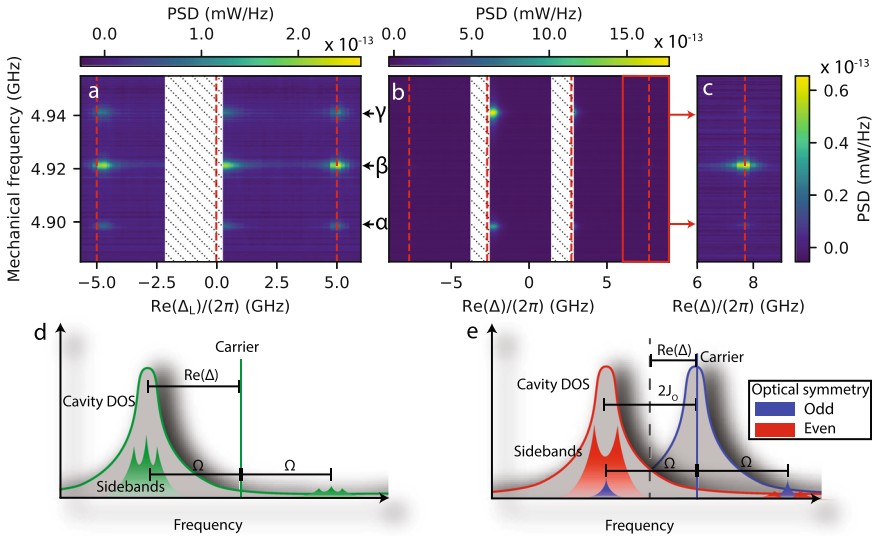

**Fig. 3 | Linear transduction of mechanical modes with different symmetries.**
Panels **a**–**c** show spectrograms of photocurrent PSD for various laser-cavity
detunings Δ normalised to 1 μW input power. **a** Spectrogram for detuned cavities.
**b** Spectrogram for a tuned cavity, with **c** an additional narrow sweep with more
optical power, scaled to match data in (**b**). Hatched regions indicate Δ values not
accessible due to thermo-optical bistability. The vertical dashed lines represent
resonant detunings, where either carrier or sideband resonates with an optical
mode. Optical input powers used: 1.3 μW (**a**), 1.5 μW (**b**) and 6.8 μW (**c**). Panels d and
e are diagrams illustrating the different frequency components and resonance
conditions. **d** Diagram for detuned devices and $\mathrm{Re}(\Delta_L)/(2\pi) = \Omega/(2\pi) \approx 4.9$ GHz,
showing the sideband resonance condition from single-mode optomechanics.
**e** Diagram for tuned devices and $\mathrm{Re}(\Delta)/(2\pi) = J_O/(2\pi) \approx 2.7$ GHz, showing
enhancement per input power for odd mechanical modes. DOS: density of states.

which we label $\alpha$, $\beta$ and $\gamma$ in order of increasing frequency. For all
modes, the transduced signal peaks at $\mathrm{Re}(\Delta_L) = -\Omega$ and $\mathrm{Re}(\Delta_L) = \Omega_m$,
corresponding to the resonance condition for the upper and lower
sideband, respectively (as per Fig. 3d). In addition, transduction for
these modes also peaks when approaching laser resonance
($\mathrm{Re}(\Delta_L) = 0$) from the blue side. For the hatched area, which contains
the exact condition of laser resonance, no data could be taken, as for
the required input power ($\approx 1$ μW) these values of Δ are not reachable
due to thermo-optical bistability.

By performing these measurements via both the right and left
waveguide, interrogating the right and left localised optical modes, we
find that these mechanical modes are present in both optical cavities
with comparable coupling strengths, showing that the mechanical
modes are delocalised supermodes. In a perfect system, these super-
modes have either an odd or even symmetry between the left and right
halves of the device. We calibrate the vacuum optomechanical coupling
rates $g_{i,j}$ ($i \in \{L, R\}$, $j \in \{\alpha, \beta, \gamma\}$) using frequency noise calibration[37]
(see 'Methods' section for more details). We find that $|g_{L,j}|/(2\pi) =$
(229, 517, 505) kHz and $|g_{R,j}|/(2\pi) =$ (375, 608, 443) kHz.

Next, in Fig. 3b, we perform similar measurements for the tuned
system. Note that for this configuration, $\mathrm{Re}(\Delta_L) = \mathrm{Re}(\Delta_R) = \mathrm{Re}(\Delta)$ and
$\mathrm{Re}(\Delta) = 0$ when the measurement laser is exactly between the two
optical supermodes. Strikingly, the transduction is now largest when
the laser approaches one of the two supermodes ($\mathrm{Re}(\Delta) = \pm J_O$). Here,
importantly, only modes $\alpha$ and $\gamma$ are visible. Figure 3c shows an addi-
tional dataset taken around sideband resonance $\mathrm{Re}(\Delta) = J_O + \Omega$ with
more optical power, showing that, conversely, for this detuning, only
mode $\beta$ is significantly transduced onto the optical field. Note that
there are now two hatched regions, corresponding to the inaccessible
red flanks of the two optical supermodes.

The difference in transduction between modes $\alpha$ and $\gamma$ and $\beta$ is
determined by the symmetry of the modes. As explained below Eq. (2),
odd mechanical modes create a cross-mode coupling, meaning that
they create sidebands in the optical mode with opposite symmetry to
that of the carrier light, whereas even mechanical modes create self-
mode coupling and thus sidebands with the same symmetry. The data
in Fig. 3b and c suggests that mechanical modes $\alpha$ and $\gamma$ are

predominantly odd, whilst mode $\beta$ is even. Figure 3e illustrates this
argument: the carrier, predominantly exciting the even mode, has
sidebands of modes $\alpha$, $\beta$ and $\gamma$ that have the same frequency as the odd
optical mode. However, only the sidebands scattered from modes $\alpha$
and $\gamma$ have the odd optical symmetry and are thus resonantly
enhanced. The mode $\beta$ sideband is effectively off-resonance. In fact,
modes $\alpha$ and $\gamma$ experience the enhancement of linear transduction
mentioned in the introduction, which we discuss in more detail in the
following.

## Enhanced linear transduction in a coupled-mode system

We now quantify the strength of the transduction signal by using
heterodyne detection. This allows for better signal-to-noise ratio and
for quantitative comparison between linear and nonlinear transduc-
tion later on. Using this setup, we perform narrow sweeps around
optimum laser-cavity detunings in detuned and tuned systems. For
each sweep, the trace with the largest transduction is plotted in Fig. 4a,
normalised to 1 μW of input power. By keeping track of other experi-
mental parameters, direct quantitative comparison between traces is
possible (see 'Methods' for more details).

We compare the optimum transduction per input power for a
detuned cavity (blue data, $\mathrm{Re}(\Delta_L) = \Omega$) to that of a tuned cavity (purple
data, $\mathrm{Re}(\Delta) = J_O$) and see a clear enhancement for odd mechanical
modes $\alpha$ and $\gamma$, but suppression of even mode $\beta$, just as in Fig. 3. The
enhancement of mode $\gamma$ is stronger than for $\alpha$, which is expected as
mode $\alpha$ also has a significant component of even symmetry, as can be
seen from the different magnitudes of $g_{L,\alpha}$ and $g_{R,\alpha}$. We also show a
trace for tuned cavities with a different detuning (green data,
$\mathrm{Re}(\Delta) = J_O + \Omega$), which exhibits the opposite effect: a stronger sup-
pression of modes $\alpha$ and $\gamma$ than of mode $\beta$, which can also be under-
stood by comparing the sideband frequencies and symmetries to the
optical modes.

These effects can be explained using the theoretical model
described above. From Eq. (5) for an odd mechanical mode in a tuned
system, it can be seen that transduction can be optimised for
$\mathrm{Re}(\Delta) = J_O$, $J_O = \Omega/2$. Both terms in the denominator are minimised
simultaneously, which can be interpreted as resonance of both the

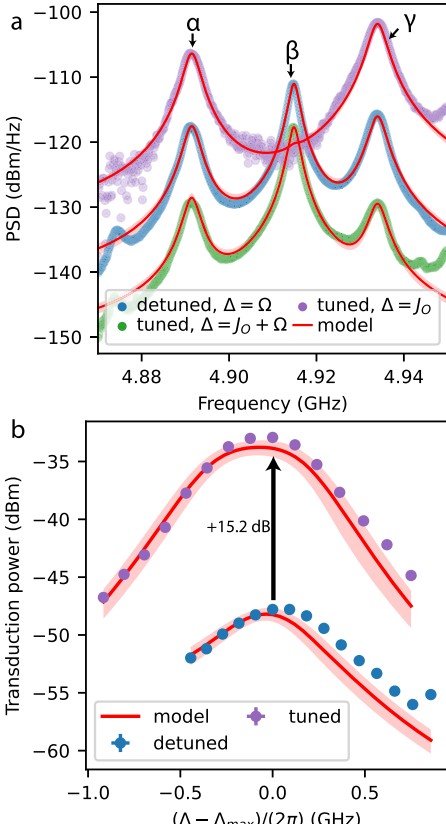

**Fig. 4 | Enhanced linear transduction in coupled-mode devices. a** Photocurrent power spectral density (PSD) per µW input power for detuned and tuned devices for several interesting laser-cavity detunings Δ around the optimum value $\Delta_{max}$, showing enhancement of transduction for tuned devices. Red curve shows the model scaled only to the blue trace, with the highlighted area displaying possible variation due to parameter uncertainty. Input powers used: 36 µW (blue), 68 µW (green), 109 nW (purple), exact values of $Re(\Delta_{max})/(2\pi)$: 5.00 GHz (blue), 7.61 GHz (green), 2.45 GHz (purple). **b** The fitted area of mode γ for detuned and tuned cavities, varying Δ around the optimum value. Vertical error bars are standard deviations in the area due to fit uncertainty and input power uncertainty. Horizontal error bars are given by standard deviation of the cavity resonance frequency fit. Both are smaller than the marker size.

input light and a sideband, as in Fig. 3e. Such simultaneous resonance is not possible if the mechanical mode has even symmetry or if the optical modes are detuned and thus the coupled-mode system shows transduction that is enhanced with respect to a single mode system. The full equations of our model (see 'Methods' section) are plotted as red lines alongside the data in Fig. 4. The model uses as parameters the independently measured optical parameters ($\kappa_i$, $\kappa_{ex,L}$, $J_O$), OM coupling constants ($g_{i,j}$) and the mechanical frequencies and linewidths extracted from a fit of the detuned cavity data with low optical power. The model prediction is scaled overall to the detuned cavity data (blue) to find the unknown photodetector conversion factor, but after that gives completely independent predictions for the tuned cavity transduction. We find excellent agreement between the model and measured data, further strengthening our conclusion about the symmetries of the mechanical modes and the enhancement mechanism at play. The shaded areas around the red lines indicate possible variation in the predicted trend due to uncertainty in the input parameters.

Finally, we quantify the enhancement of transduction for mode γ by fitting the total area of the signal, giving us the total transduced power. In Fig. 4b, we plot this power as a function of the laser-cavity detuning around the optimum point for tuned and detuned cavities. We find an enhancement of a factor of 32.9 ± 0.5 (+15.2 dB) between

the optimal transduced powers, where uncertainty is dominated by error on the measurement of the low input power. Note that this enhancement is for constant input power to both the detuned and tuned system. Again, the red lines give the powers extracted from the model, using independent parameters and using the same scaling as in Fig. 4a. We compare this enhancement to the ideal value of $(\Omega/\kappa)^2 \approx 56$. As confirmed by numerical calculations, the lower realised enhancement can be explained well by the non-ideal value of $J_O$, namely that $J_O - \Omega/2 \approx 0.8\kappa$.

## Multimode enhancement of optomechanical nonlinearity

Nonlinear transduction of mechanical motion is detectable as optical fluctuations at twice the mechanical frequency. In the weak coupling regime, nonlinear sidebands can be viewed as being created by sequential scattering of light from first-order sidebands[17]. This process can involve the same mechanical mode twice, or combine different modes, and results in fluctuations at $\Omega_j + \Omega_k$, where $j,k \in \{\alpha, \beta, \gamma\}$. Classically, a nonlinear sideband contains information about $x_j x_k$, the position-product of mechanical modes $j, k$ creating the sideband. In the quantum regime, a sideband-resolved system does not allow detection of position-squared, as can be seen by filling in $\hat{x} = x_{zpf}(\hat{b}^\dagger + \hat{b})$ in Eq. (3): not all resulting terms will be resonant simultaneously[11]. Instead, for on-resonant driving, the optical frequency is determined by the phonon number $\hat{b}^\dagger \hat{b}$, in principle allowing for QND measurement of phonon number. The resulting signal will be centred around zero frequency, making it very hard to detect in our setup. In our experiment, we detect second-order sidebands at $-2\Omega_m$ away from the carrier, which, in terms of the Hamiltonian, corresponds to the term $\propto \hat{b}^\dagger \hat{b}^\dagger$ that can be used for mechanical squeezing[38] and heralded two-phonon generation[17].

In Fig. 5a and b, we show the photocurrent spectra of nonlinear transduction in detuned (a) and tuned cavities (b) for optimal detuning ($Re(\Delta_L) = \Omega$ and $Re(\Delta) = J_O + \Omega$, respectively). The spectra are normalized to one intracavity photon, isolating enhancement of the nonlinear optomechanical processes inside the device from input resonance effects, such as the linear enhancement. Moreover, the intracavity photon number is the limiting factor in many experiments due to heating[7,26,31]. Normalising to input power instead would reduce the tuned cavity PSD by a factor of roughly 2.

The nonlinear spectra contain several peaks, which can all be attributed to a specific mixing of two mechanical modes by matching the frequency to the sum frequency of two linear transduction peaks (see Fig. 5a and b). There is a clear enhancement of signal from several nonlinear scattering processes, most notably $\beta + \gamma$, $\alpha + \beta$ and $2\gamma$. For an intuitive understanding of the relative strength of these processes, we have a closer look at the largest peak, $\beta + \gamma$. In Fig. 5c, we depict schematically how this particular scattering achieves the optimal resonance condition. The carrier light, exciting mostly the odd optical mode, is resonantly scattered into the odd mode through even mechanical mode $\beta$, and subsequently scattered resonantly by odd mode $\gamma$ into the even (opposite symmetry) optical mode. As such, the process is resonant and symmetry-conserving, ensuring maximal enhancement. The $2\gamma$ process is also enhanced and can be described by the simplified transduction expression in Eq. (9). For $Re(\Delta) = J_O + \Omega_\gamma$ and $J_O = \Omega_\gamma/2$, both terms in the denominator are minimized, which can be interpreted as simultaneous resonance of first and second sidebands, and transduction is enhanced over a single-mode device. The $2\gamma$ peak is less strong than the $\beta + \gamma$ peak, as the former requires carrier occupation of the even optical mode, which is further detuned from the laser.

We compare our experimental results to the model for nonlinear transduction. Note that this model gives an independent prediction, as it is calculated with independently measured system parameters and scaled only once to linear transduction data of the detuned system. In Fig. 5, we plot this model as a red line. We see that the different

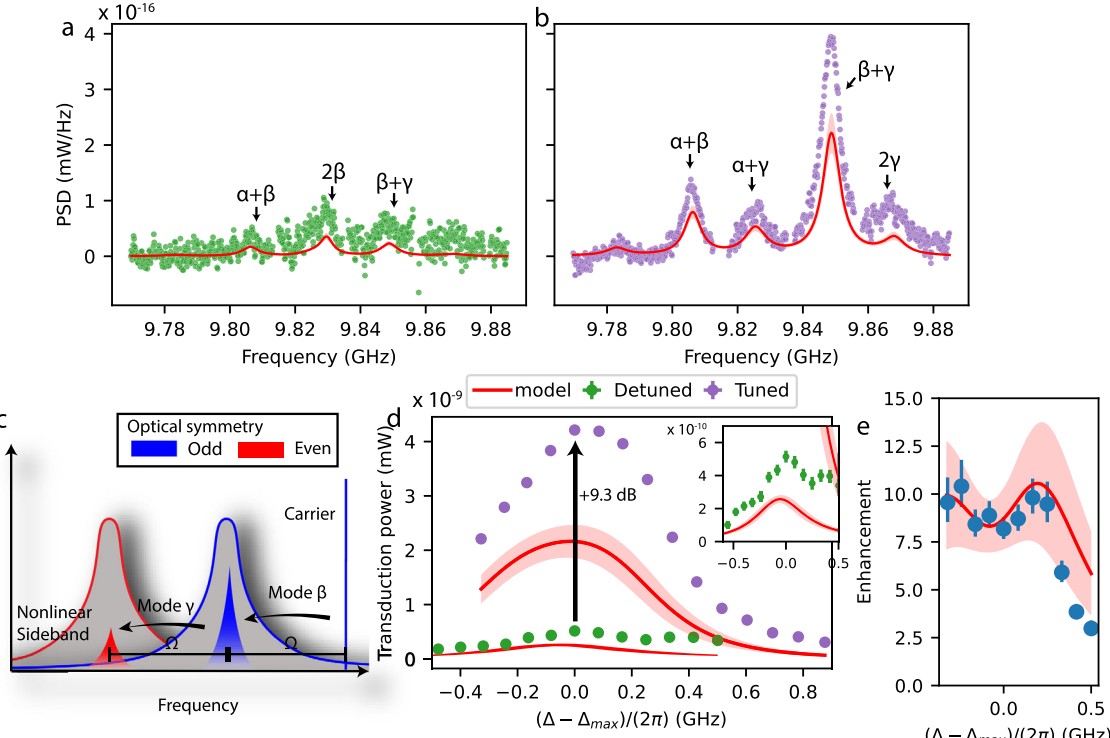

**Fig. 5 | Enhanced nonlinear transduction.** In panels **a** and **b**, we compare the optimal nonlinear part of the power spectral density (PSD) for detuned (**a**) to tuned (**b**) modes. PSD was normalised to per cavity photon. The red line is the independent model prediction, the shaded region is the estimated uncertainty on the model. The detuning (power used) was 5.15 GHz (69 μW) and 7.52 GHz (152 μW) for (**a**) and (**b**), respectively. **c** A schematic representation of the optimal resonance condition for scattering from mode $\beta$ and then mode $\gamma$. **d** The fitted area of the $\beta + \gamma$ tone for detuned and tuned modes while varying $\Delta$ around the optimal point. The inset is a zoom-in of the detuned cavity data. Vertical error bars are standard deviations in area due to fit uncertainty, horizontal error bars are standard deviation in measurement of $\Delta$. **e** The enhancement of nonlinear transduction expressed as transduction power of mode $\beta + \gamma$ for a tuned system divided by that of a detuned system (the purple and green data in panel d, respectively), around optimum detuning $\Delta_{max}$. To do this, the purple data was interpolated. As before, the red line with shaded area shows the model value and corresponding uncertainty. The vertical error bars correspond to propagated standard deviation from fit uncertainties in area.

nonlinear transduction peaks and their relative sizes are captured well by the model. Overall, the model is found to predict a smaller signal than measured experimentally, with a difference larger than expected based on statistic uncertainty in the estimated system parameters. We provide details on those estimations in Supplementary Notes 8 and 9. The deviation of the absolute transduced signal from the model must thus be due to a systematic error, as we discuss further in the Discussion section. Notably, we observe that the deviation affects both the single-cavity and coupled-cavity measurements.

To quantify the degree of enhancement due to multimode interactions, we compare the total power of scattering process $\beta + \gamma$ for detuned and tuned cavities. In Fig. 5d, we plot the fitted areas and the model for a sweep of laser-cavity detuning $\Delta$ around the optimal value $\Delta_{max}$. We find an enhancement of a factor $8.4 \pm 0.6$ (+9.3 dB), demonstrating enhanced nonlinear optomechanical processes in a coupled-mode device by direct comparison to a single-mode configuration in the same device. The full model, based on the fitted device parameters, predicts an enhancement at resonant detuning of a factor 8.7, in good agreement with the experimental data obtained by dividing the data of single and coupled-cavity device configurations. In Fig. 5e, we show the enhancement factor, defined as the ratio in transduced powers between the tuned and detuned cavity case for varying laser-cavity detuning around the respective optimal values. In addition, we plot the model prediction for this enhancement. We find that the model predicts the enhancement factor well, showing that it is able to predict correctly how nonlinear transduction changes between the tuned and detuned configuration, as well as for different values of laser-cavity detuning.

We note that the ideal theory for optimally detuned coupling predicted an enhancement of $2\left(\frac{\Omega}{\kappa}\right)^2$ for nonlinear scattering from a single mechanical mode. For this particular scattering $\beta + \gamma$ with two different modes, nonlinear sideband power in the tuned system is reduced by a factor 4, as only scattering from $\beta$ then $\gamma$ is enhanced, while scattering from $\gamma$ then $\beta$ is off-resonant. In the detuned case, both processes have equal amplitude, which means the expected enhancement is $\frac{1}{2}\left(\frac{\Omega}{\kappa}\right)^2 \approx 28$. We have confirmed numerically that this is a good approximation for a system with $2J_O = \Omega$, and that our lower observed enhancement can be explained through the non-ideal value of $J_O$.

## Discussion

We have demonstrated in direct comparison an 8-fold enhanced nonlinear transduction in our coupled mode system, as well as a 33-fold enhanced linear transduction with respect to input power. This demonstration confirms experimentally the idea that optomechanical nonlinearity can be enhanced in a sideband-resolved coupled-mode system. The enhancement was determined by using two configurations of the same device, either tuning the two optical modes to the same frequency, or detuning one completely, effectively removing it from the system and leaving a single-mode device with the same parameters. The mechanical modes remain delocalised, giving multiple mechanical modes even in the single-cavity configuration, at the cost of the vacuum optomechanical coupling $g_0$ being reduced by $\sqrt{2}$ from an uncoupled single cavity, due to the increased mass of the modes.

In addition, we have provided a theoretical framework that predicts the linear and nonlinear transduction based on independently

measured optical, mechanical and optomechanical parameters. The observed enhancement in the coupled-cavity configuration is explained well by the model. A discrepancy exists between the overall ratio of linear to nonlinear transduction in the data and the model as seen in Fig. 5d, leading to an underestimation of nonlinear transduction by the model. An error in the determination of the model parameters could affect this predicted ratio. In Supplementary Notes 8 and 9, we provide an extensive discussion of the determination of vacuum optomechanical coupling rates $g_{L(R),j}$, the optical linewidths $\kappa_{L(R)}$ and cavity-waveguide outcoupling rates $\kappa_{ex,L(R)}$. The estimated uncertainties on these parameters were used to determine the error region on the model in Figs. 4 and 5, from which we conclude that these uncertainties are not sufficient to explain the overall scaling difference between the model and data for nonlinear transduction.

Another relevant model parameter is the amplitude of thermomechanical motion, which we assume implicitly to be constant between measurements. This motion is determined by, among other factors, the effective mechanical decay rate and the temperature of the environment. As we use thermal tuning, we do affect the mode environment temperature slightly, although the estimated temperature increase is only 6.4 K (see Supplementary Note 4 for details), which is marginal compared to the base temperature of 293 K and would lead to an overestimation of nonlinear transduction by 4%. Therefore, we neglect the effect of this temperature increase in analysis. Next, the effective mechanical decay rate could be changed through the optomechanical interaction via dynamical backaction. In our model, we have included this effect (see 'Methods' section). At the same time, care was taken to keep dynamical backaction effects small during the experiments described here. Still, these effects alter the mechanical position variance slightly, most prominently for mode $\beta$. For the linear enhancement measurement we estimate $\langle x_\beta^2 \rangle_{tuned} / \langle x_\beta^2 \rangle_{detuned} \approx 0.93$ and for the nonlinear enhancement measurement we estimate $\langle x_\beta^2 \rangle_{tuned} / \langle x_\beta^2 \rangle_{detuned} \approx 0.96$. The enhancement factors given in the results section have been compensated for this small effect of dynamical backaction.

Together, the above considerations do not yet fully explain the discrepancy between nonlinear transduction in the data and model. In further research, it may be of use to investigate the effect of photothermal effects on the system[39] which may cause unexpected dynamical backaction effects, and to study the transduced signal as a function of environmental temperature. With reduced dissipation at cryogenic temperatures it could also be possible to use dynamical backaction instabilities to estimate optomechanical coupling rates, thus providing independent verification of their magnitude and the expected nonlinearity[40]. Also, it might be possible that strong optical fields induce correlations between different mechanical modes[41], which could potentially affect the ratio of nonlinear to linear transduction.

For the coupled-mode device, we identified optimal enhancement values of optomechanically scattered powers of $\mathcal{E}^{lin} = \left(\frac{\Omega}{\kappa}\right)^2$ and $\mathcal{E}^{qua} = 2\left(\frac{\Omega}{\kappa}\right)^2$ for linear and nonlinear transduction, respectively. These enhancement factors could, however, both be increased to $\left(2\frac{\Omega}{\kappa}\right)^2$. Tuning the system from effectively single-mode to coupled-mode reduces the effective coupling rate of the optical eigenmodes with the outcoupling waveguide, resulting in less cavity photons and smaller cavity-to-detector efficiency thus giving the lower theoretical maxima we find. This can be overcome by designing the individual cavities with a larger $\kappa_{ex,L(R)}$. Next, when comparing the experimentally found enhancement to these theoretical values, we find a deviation because of the non-ideal optical coupling rate. A further fine tuning of the optical coupling rate $J_O$ will allow the device performance to approach optimal enhancement. We also note that some nonlinear scattering processes can be selected by optical excitation of only one particular supermode, which can be achieved by exciting via both on-chip waveguides simultaneously and (anti)symmetrically. Altogether, the maximal nonlinear enhancement of $(2\Omega/\kappa)^2 \approx 225$ can be approached in

this device by simple redesign within existing possibilities, without the need to further increase optical quality factor.

We have identified the fabrication angle between the device and silicon crystal axis as a degree of freedom to control the mechanical properties of the device, without affecting optical properties. Effectively, additional inter-cavity coupling $J_M$ was created by leveraging this angle to introduce a new cavity decay channel. Although this, in principle, also increases mechanical radiative decay into the substrate and thus decreases mechanical quality factor, such decrease had only limited effect on our experiment, as these mechanical modes are limited by non-radiative decay channels at room temperature[42]. Moving forward, to recover the mechanical quality factor, cryogenic operation would be crucial. In addition, one could terminate nanobeam ends into a structure that has a full phononic bandgap[32] to lower radiative decay. Alternatively, further optimisation of coupling region to optimize $J_M$ and $J_O$ simultaneously without using anisotropy could be performed.

Optical post-fabrication tuning was performed through thermal tuning with an auxiliary laser and a thermally isolated device design. The main advantages of this method are the accuracy, reversibility and ease of use. Although temperature increase in the device is only a few Kelvin, the requirement of constant heating can possibly be difficult in cryogenic conditions. Moving forward, it would thus be highly opportune to investigate replacing the tuning method by other methods that are compatible with cryogenics and quantum experiments, such as oxidation tuning[43,44], light-induced chemical etching[45] or laser-induced gas desorption[46].

Looking ahead, the coupled-mode system presented here has several applications, both in the classical and in the quantum regime. First, two optical supermodes at specific frequency separation, combined with several closely spaced mechanical modes provide a very interesting platform for studying mechanical lasing in multiple modes and optomechanical frequency combs[47,48], for which no further device improvements are necessary. In particular, multiple optical resonances allow for the resonant enhancement of specific frequencies from frequency combs, allowing for selective frequency multiplication. For many different applications, our design can be applied to reduce the input power for optical measurement, especially useful in cryogenic applications involving superconducting circuits next to optical components, where optical absorption can degrade performance[7]. Still in the weak coupling regime, non-classical states can be generated by heralding, and our device can be used for heralded creation of two-phonon states[17]. Moreover, the enhanced nonlinearity could be used to reveal the granularity of mechanical energy by detecting phonon shot noise[12].

Importantly, in the weak coupling regime, a strong linear coupling persists next to the nonlinear coupling, which can be an important factor of decoherence in the generation of non-classical mechanical states[17,29]. One possible method to mitigate the effect of linear decoherence and allow for measurement-based non-classical state generation is the use of feedback[9]. Alternatively, with further improvements to the vacuum coupling rate $g_0$ and reduction of optical decay rate $\kappa$, approaching the SPSC regime, other quantum applications will come in reach more quickly by use of the coupled-mode device presented here, even in the presence of linear decoherence. These include the photon blockade effect for the deterministic generation of single-photon states[14] and phonon number measurements[16].

## Methods

### Device fabrication
Devices were fabricated from a silicon-on-insulator wafer with a 220 nm Si device layer on top of a 3 μm $SiO_2$ sacrificial layer. The Si device layer follows the (100) crystal plane and devices were fabricated at an angle of $\theta = 15°$ to the $\langle 010 \rangle$ axis. E-beam exposure was used to pattern an HSQ resist layer, followed by development in 25% TMAH. Anisotropic plasma etching was performed using a

mixture of HBr, $O_2$ and $Cl_2$ to etch the silicon device layer. Finally, the $SiO_2$ layer was removed using a 40% HF etch. After this etch, the device is transported to the setup vacuum chamber within half an hour to prevent oxidation of the Si surface.

## Direct detection setup

The sample is placed in a vacuum chamber which is pumped down and filled with nitrogen back to 0.25 bar to prevent oxidation of the nanobeam surface. Optical connection to the sample was made via a dimpled optical fibre[49]. Light from a tunable diode laser (Toptica CTL 1500) was sent into the nanobeam and upon reflection was amplified in an erbium-doped fibre amplifier (EDFA, Calmar Coronado) and detected on a 12 GHz photodiode (New Focus 1544-B). The photocurrent was analysed on a real-time spectrum analyser (Agilent MXA N9020A). To keep track of the optical modes, an additional tunable laser (New Focus TLB-6728) was swept across the optical modes intermittently with low optical power of ≈100 nW. The laser was modulated strongly at 1 MHz and a measure of reflection was obtained with a lock-in measurement of reflected power to overcome detector electronic noise. Determination of optomechanical vacuum coupling rate was done using frequency noise calibration with an electro-optical phase modulator calibrated using a fibre-loop cavity[37,50], of which details are discussed in Supplementary Note 8. In Supplementary Notes 1, 2 and 3 and Supplementary Fig. 1 we describe the further details of the experimental setups.

## Heterodyne setup

For the heterodyne setup, an additional Toptica CTL tunable diode laser was used as a local oscillator (LO). The two Toptica lasers were locked at a fixed frequency offset by creating sidebands on one laser using an electro-optic modulator and locking the other laser to this sideband. The lock was achieved using a Red Pitaya digital signal processor, applying feedback to the diode current and tuning piezo of the LO laser, and the resulting beating between the two lasers has a linewidth much smaller than the mechanical linewidth. To be able to quantitatively compare different measurements, care was taken to keep constant the LO power, as well as the polarisation overlap between the two lasers and the dimple-to-waveguide coupling efficiency.

## Coupled-mode model

To derive a model that can predict the photocurrent based on parameters of the mechanical and optical modes, we start with the equations-of-motion (EOMs) for the classical optical field amplitudes of left and right optical modes and mechanical mode displacements of modes $\alpha, \beta, \gamma$ in a frame rotating at optical input frequency $\omega_{in}$[51]

$$\dot{a}_{L(R)} = i\left(\Delta_{L(R)} + \sum_{j=\alpha,\beta,\gamma} g_{L,j} x_j\right) a_{L(R)} + i J_O a_{R(L)} + \sqrt{\kappa_{ex,L(R)}} a_{in,L(R)}, \quad (12)$$

$$\ddot{x}_j + \Gamma_j \dot{x} + \Omega_j^2 x = 2\sqrt{\Gamma_j \Omega_j} p_{in,j} + 2\left(g_{L,j}|a_L|^2 + g_{R,j}|a_R|^2\right), \quad (13)$$

where $\Delta_{L(R)} = (\omega_{in} - \omega_{L(R)}) + i\kappa_{L(R)}/2$ contains both detuning between input field and the optical mode frequencies $\omega_{L(R)}$ and the optical decay rate $\kappa_{L(R)}$. The mechanical modes $j \in \{\alpha, \beta, \gamma\}$ have frequencies (decay rates) $\Omega_j$ ($\Gamma_j$). Inter-mode optical coupling is given by $J_O$ and optomechanical coupling is given by $g_{L(R),j}$. Position is expressed as unitless position $x_j = q_j/x_{zpf}$, where $q_j$ is the mode amplitude in metres and $x_{zpf} = \sqrt{\frac{\hbar}{2m\Omega}}$ is the zero-point amplitude of the mode, with $m$ the mode effective mass. Finally, optical modes are connected to input fields $a_{in,L(R)}$ and mechanical modes to thermal bath momenta $p_{in,j}$.

The EOMs are solved in a perturbative fashion[17], $a_{L(R)}(t) = \bar{a}_{L(R)} + a_{L(R)}^{(1)}(t) + a_{L(R)}^{(2)}(t) + ...$, where $\bar{a}_{L(R)}$ is the steady-state cavity field and $a_{L(R)}^i$ contains all terms of $i$-th order in $g_{L(R),j}$. This requires that thermomechanical motion is sufficiently small, i.e. that $g_{L(R),j}\sqrt{n_{th,j}}$, with thermal phonon occupation $n_{th,j} = k_B T/(\hbar\Omega_j)$, $k_B$ being the Boltzmann constant and $T$ temperature, is smaller than the optical linewidth $\kappa$. For our system, $g_{L(R),j}\sqrt{n_{th,j}}/\kappa \approx 0.04$ and we are thus well in the perturbative regime. Also, we assume we connect optically to the left cavity, i.e. $\kappa_{ex,R} = 0$.

Solving is done in the frequency domain, for which we use the Fourier transform

$$A[\omega] = \int_{-\infty}^{\infty} a(t) e^{i\omega t} dt. \quad (14)$$

To transform a product of functions, we use the following identity:

$$
\begin{aligned}
(AB)[\omega] &= \int_{-\infty}^{\infty} b(t)a(t) e^{i\omega t} dt \\
&= 1/(2\pi) \int_{-\infty}^{\infty} A[\omega']B[\omega - \omega']d\omega' \\
&= 1/(2\pi)A[\omega]*B[\omega].
\end{aligned}
\quad (15)
$$

We find

$$\bar{a}_L = \frac{i\sqrt{\kappa_{ex,L}}\Delta_R}{\Delta_L \Delta_R - J_O^2} \bar{a}_{in,L}, \quad (16)$$

$$\bar{a}_R = \frac{-iJ_O\sqrt{\kappa_{ex,R}}}{\Delta_L \Delta_R - J_O^2} \bar{a}_{in,L}, \quad (17)$$

$$
\begin{aligned}
A_{L(R)}^{(1)}[\omega] &= \sum_{j=\alpha,\beta,\gamma} \frac{J_O g_{R(L),j}\bar{a}_{R(L)} - (\omega+\Delta_{R(L)})g_{L(R),j}\bar{a}_{L(R)}}{(\omega+\Delta_R)(\omega+\Delta_L) - J_O^2} X_j[\omega] \\
&= \sum_{j=\alpha,\beta,\gamma} \widetilde{M}_{L(R),j}(\omega)X_j[\omega],
\end{aligned}
\quad (18)
$$

$$
\begin{aligned}
A_L^{(2)}[\omega] = \frac{1}{2\pi} \frac{1}{(\omega+\Delta_R)(\omega+\Delta_L) - J_O^2} \sum_{k=\alpha,\beta,\gamma} \Big( & J_O g_{R,k}(A_R^{(1)}[\omega]*X_k[\omega]) \\
& - (\omega+\Delta_R)g_{L,k}(A_L^{(1)}[\omega]*X_k[\omega]) \Big),
\end{aligned}
\quad (19)
$$

where $X_j$ is the Fourier transform of mechanical displacement $x_j$ and

$$\widetilde{M}_{L(R),j} = \frac{J_O g_{R(L),j}\bar{a}_{R(L)} - (\omega+\Delta_{R(L)})g_{L(R),j}\bar{a}_{L(R)}}{(\omega+\Delta_R)(\omega+\Delta_L) - J_O^2}. \quad (20)$$

We are interested in the power spectral density (PSD) of photocurrent $I$, which is given by[52]

$$S_{II}[\omega] = \frac{1}{2\pi} \int_{-\infty}^{\infty} \langle I[\omega]I[\omega']\rangle d\omega'. \quad (21)$$

The photocurrent is equal to the optical power (removing the proportionality constant), which for the heterodyne detection is given by

$$I[\omega] = \sqrt{n_{het}} \sqrt{\kappa_{ex,L}} \left(A_{out}^{(i)}[\omega_-] + (A_{out}^{(i)}[-\omega_+])^*\right), \quad (22)$$

with $\omega_- = \omega - \omega_{het}$ and $\omega_+ = \omega + \omega_{het}$, $\omega_{het}$ the heterodyne frequency and $n_{het}$ the amount of photons in the LO. The cavity reflected light for non-zero $\omega$ is given by the input-output relation $A_{out}^{(i)} = \sqrt{\kappa_{ex,L}}A_L^{(i)}$.

Now, we need to specify the motion of $X_j[\omega]$. Following Bowen and Milburn[52], we define this to be

$$X_j[\omega] = \chi_j(\omega) P_{\mathrm{in},j}[\omega], \tag{23}$$

where $\chi_j(\omega)$ is the susceptibility of mechanical mode $j$, with first-order dynamical backaction correction

$$
\begin{aligned}
\chi_j(\omega) = 2\sqrt{\Gamma_j}\Omega_j \Big[ \Omega_j^2 - \omega^2 - i\omega\Gamma_j - 2\Omega_j \\
\Big( g_{\mathrm{L},j}\bar{a}_\mathrm{L}\widetilde{M}_{\mathrm{L},j}^*(-\omega) + g_{\mathrm{L},j}\bar{a}_\mathrm{L}^*\widetilde{M}_{\mathrm{L},j}(\omega) \\
g_{\mathrm{R},j}\bar{a}_\mathrm{R}\widetilde{M}_{\mathrm{R},j}^*(-\omega) + g_{\mathrm{R},j}\bar{a}_\mathrm{R}^*\widetilde{M}_{\mathrm{R},j}(\omega) \Big) \Big]^{-1},
\end{aligned}
\tag{24}
$$

and $P_{\mathrm{in},j}$ the thermal bath forcing (momentum) term, which is a white noise with correlation function

$$\langle P_{\mathrm{in},j}[\omega] P_{\mathrm{in},k}[\omega'] \rangle = 2\pi n_{\mathrm{th}} \delta_{j,k} \delta(\omega+\omega'). \tag{25}$$

In the calculation of second-order PSD, the correlation function of a product of four thermal bath momenta has to be calculated. To evaluate this, we use the fact that, in thermal equilibrium, the momenta are normally distributed to employ the Isserlis-Wick theorem[9]. In particular, the expectation value of the product of four normally-distributed random variables $Y_l$ can be reduced to

$$\langle Y_1 Y_2 Y_3 Y_4 \rangle = \langle Y_1 Y_2 \rangle \langle Y_3 Y_4 \rangle + \langle Y_1 Y_3 \rangle \langle Y_2 Y_4 \rangle + \langle Y_1 Y_4 \rangle \langle Y_2 Y_3 \rangle. \tag{26}$$

Combining all of the previous steps, we can write down expressions for the first and second-order components of $S_\mathrm{II}$

$$
\begin{aligned}
S_\mathrm{II}^{(1)}[\omega] = \kappa_{\mathrm{ex,L}} n_{\mathrm{het}} n_{\mathrm{th}} \sum_j \Big[ M_j(\omega_-) M_j(-\omega_-) \\
+ M_j(\omega_-)(M_j(\omega_-))^* (M_j(-\omega_+))^* M_j(-\omega_+) \\
+ (M_j(-\omega_+))^* (M_j(\omega_+))^* \Big],
\end{aligned}
\tag{27}
$$

with

$$M_j(\omega) = M_{\mathrm{L},j}(\omega) = \widetilde{M}_{\mathrm{L},j}(\omega)\chi_j(\omega), \tag{28}$$

and we remember that our experiment only probes the left cavity L. Note that $M_{\mathrm{R},j}$ is obtained from $M_{\mathrm{L},j}$ by swapping subscripts R and L. For nonlinear transduction, we find

$$
\begin{aligned}
S_\mathrm{II}^{(2)}[\omega] = \frac{\kappa_{\mathrm{ex,L}} n_{\mathrm{het}} n_{\mathrm{th}}^2}{2\pi} \int d\omega' \sum_{j,k=\alpha,\beta,\gamma} \Big( N_{j,k}(\omega_-,\omega') \\
\Big[ N_{j,k}(-\omega_-,-\omega') + N_{k,j}(-\omega_-,\omega'-\omega_-) \\
+ N_{j,k}^*(\omega_-,\omega') + N_{k,j}^*(\omega_-,\omega_--\omega') \Big] \\
+ N_{j,k}^*(-\omega_+,\omega') \Big[ N_{j,k}(-\omega_+,\omega') \\
+ N_{k,j}(-\omega_+,-\omega_+-\omega') + N_{j,k}^*(\omega_+,-\omega') \\
+ N_{k,j}^*(\omega_+,\omega'+\omega_+) \Big] \Big),
\end{aligned}
\tag{29}
$$

with

$$
\begin{aligned}
N_{j,k}(\omega,\omega') = \frac{1}{(\omega+\Delta_\mathrm{R})(\omega+\Delta_\mathrm{L}) - J_\mathrm{O}^2} \Big( J_\mathrm{O} g_{\mathrm{R},k} M_{\mathrm{R},j}(\omega-\omega') \\
- (\omega+\Delta_\mathrm{R}) g_{\mathrm{L},k} M_{\mathrm{L},j}(\omega-\omega') \Big) \chi_k(\omega').
\end{aligned}
\tag{30}
$$

## Data availability

The data in this study are available from the Zenodo repository at https://doi.org/10.5281/zenodo.7307901.

## Code availability

The code used in the present work is available from the authors upon request.

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

## Acknowledgements

We would like to thank P. Busi, A. Fiore, M. Lodde and P. Neveu for valuable discussions. We thank F. Koenderink for critical reading of the manuscript. This work is part of the research programme of the Neth-erlands Organisation for Scientific Research (NWO). E.V. acknowledges support from NWO Vidi, Projectruimte, and Vrij Programma (Grant No.680.92.18.04) Grants, and the European Research Council (ERC Starting Grant No. 759644-TOPP).

## Author contributions

R.B. and E.V. conceived the project. R.B. designed and fabricated the sample. R.B. and E.V. designed the experiment, R.B. built the setup, performed measurements, performed data analysis and derived model equations. E.V. supervised the project. Both authors participated in the writing of the manuscript.

## Competing interests

The authors declare no competing interests.
