## [Peer Review File · Nature Communications]

Enhanced nonlinear optomechanics in a coupled-mode photonic crystal deviceREVIEWER COMMENTS

Reviewer #1 (Remarks to the Author):

The paper presents experimental results related to a coupled-mode optomechanical system, characterized by two optical modes which are coupled together and is able to enhance linear and nonlinear transduction, under specific resonance conditions, compared to the single mode case. This is known since many years from now but the experimental demonstration is lacking because of the difficulty to tune in situ the optomechanical couplings and the direct optical coupling.

The paper presents a quite clear experimental demonstration of enhanced transduction both in the linear and in the nonlinear regime and this is an important results for the field and for science in general because optomechanical sensors can be applied in many scientific and technological areas.

The paper first describes the model and the basic expressions for the power spectral density of the heterodyne detection of mechanical motion, reconsidering older results in the literature, and underlining the fact that the enhancement due to the coupled modes, when the resonance condition are satisfied, is associated with the resolved sideband ratio (mechanical frequency/cavity decay rate).

The key technical advancements allowing to achieve the novel results are: i) exploitation of the material mechanical anisotropy, by tuning the angle θ between the material 010 crystal axis and the optical axis of the device; this allows independent mechanical tuning and optical tuning of the mode properties.

ii) Post-fabrication tuning of the optical mode coupling via a tunable thermal gradient, allowing to control the optical mode splitting and coupling to be adjusted, and to have the desired coupled-mode system (Fig. 2b)

These two technical solutions are novel and relevant, and they represent the basis for the relevant experimental demonstration of enhanced mechanical detection, both in the linear and nonlinear regime, shown in Fig 4 and Fig 5.

I think that the paper deserves publication in Nature Communication because of its relevance and broad interest. Even though the presentation is quite clear and, together with the supplementary information and Methods, provides a quite complete system and methodology description, I think that there are few points that should be fixed before publication in order to improve the results and the presentation.

1. The model adopted describes very well the linear transduction data (Fig. 4), while it underestimates the effect in the nonlinear case (Fig. 5). This discrepancy is not discussed in a convincing way in the paper and I would provide a deeper discussion of it.

The perturbation theoretical treatment adopted to describe the model seems reasonable and appropriate for the given parameters, and in fact, the authors do not attribute the discrepancy to this issue.

The fact suggested by the authors that the value of the vacuum optomechanical coupling rates g_L and g_R can be wrongly estimated may be reasonable in my opinion, but I would further investigate this point. The calibration method of Ref. 36 is used for the coupling measurement, but this method assumes the knowledge of the thermal phonon number and this is in my opinion a strong limitation, and a possible reason for the inaccurate estimation of the coupling. In fact, in this case, the thermal gradient tuning adopted may change considerably the local temperature, leading in this way to a wrong estimation of g_L and g_R . The author could profit from the alternative method for the measurement of the vacuum optomechanical coupling rate shown in P. Piergentili et al., Phys. Rev. Applied 15, 034012 (2021) which does not need the knowledge of the mean thermal phonon number.

Another possible point to check is the normalization of the calibration with respect to the cavity photon number, which may be sensitive to the value of various parameters measured independently.

I have the feeling that the results could be improved and better explained by considering in more details the above aspects, making in particular a critical analysis of the measurement of the optomechanical coupling rates.

I have also two final minor points:

(i) In the perspectives the authors mention various quantum applications. I am less optimistic: can the thermal tuning method safely applied in cryogenic scenarios? The mechanical quality factors seem quite low at the moment and I fear that one needs to improve significantly the quantum cooperativity in order to approach the quantum regime. Nonetheless I agree that operating the present coupled-mode system in the quantum regime would be of extreme relevance.

(ii) In eq (3) and also in the description of the coupled mode model in the Methods section, the notation J is used rather than J_O for the optical coupling. If I understood properly, this should be only a misprint and not a novel, undefined parameter.

Reviewer #2 (Remarks to the Author):

Optomechanics studies the radiation pressure interaction between light and mechanical motion. The field has seen rapid advances in the past decade, including both optomechanical devices that enter a variety of quantum regimes and precision optomechanical sensors. A key area of important in the field is the development of new techniques to enhance the light-matter interaction strength, both linear interactions (used, for e.g., for quantum ground state cooling) and nonlinear interactions (proposed, for e.g., to allow the generation of Schrodinger cat and phonon-number states). A long-standing proposal to enhance light-matter interactions is to use a pair of coupled optical cavities. However, because very precisely defined parameters are required for this to work, it has not previously been convincingly achieved. This manuscript convincingly shows the predicted enhancement.

The manuscript is well written and extremely comprehensive, the results are convincing and carefully analyzed. As well as demonstrating a long-predicted and important phenomenon; it introduces significant technical advances, including the development of a new coupled-cavity architecture, a new method of cavity resonance frequency tuning, and the exploitation of the anisotropy of silicon to couple pairs mechanical modes. I have no hesitation in recommending publication in Nature Communications.

I have only a few comments:

1) In paragraph 2 (starting on line 32), the authors claim that the regime of $g_0 > \kappa$ has not yet been reached. This is strictly, not true. It has been reached in atom-optomechanical systems (e.g. Science, 322 235–238 (2008)).

2) Some of the conclusions of the paper based on Fig. 3 seem unsupported by data due to the inaccessible hatched parameter regions on the figure (where data taking wasn't possible due to optomechanical instability). E.g. on line 5 of pg 6 it's claimed that the signal peaks correspond to the resonant conditions for the upper sideband, laser, and lower sideband. However, the laser-resonant peak appears not to be included in the data - this is within the hatched region if I'm not mistaken. Similarly, for line 29, it appears that the peaks for $\pm \mu J_0$ may both be in the hatched region. It's important that the authors address this. I would suggest, also, including hashed vertical lines on the experimental PSDs in Fig. 3 to show the key resonance conditions.

3) In the paragraph on page 8, line 81, the authors suggest that a systematic error might explain the deviation of their nonlinear transduction data from their model. However, I would have thought that such an error this would also degrade the (very good) agreement for linear transduction?

4) The authors show enhancement of both the linear and nonlinear optomechanical interactions. However, for the nonlinear optomechanical interaction to be effective in preparing non-classical states such as phonon-number states, it is generally crucial that the linear interaction is very strongly suppressed - the linear interaction introduces decoherence and typically has a much higher magnitude than the nonlinear interaction. It would be helpful if the authors could comment in the discussion about the prospects of non-classical state-generation generation in the presence of the linear coupling. Would it be necessary to suppress this? If so, by how much, and how can it be done? If not, why not.

Response to the reviewers

R. Burgwal and E. Verhagen,

Enhanced nonlinear optomechanics in a coupled-mode photonic crystal device

We thank both reviewers for their careful reading and thoughtful, constructive comments. We are very pleased that both reviewers recommend publication in Nature Communications. We have carefully considered their comments which have allowed us to significantly strengthen our manuscript. We address their comments (written in blue) in full below, and list the actions taken in the revised manuscript.

Reviewer #1

The paper presents experimental results related to a coupled-mode optomechanical system, characterized by two optical modes which are coupled together and is able to enhance linear and nonlinear transduction, under specific resonance conditions, compared to the single mode case. This is known since many years from now but the experimental demonstration is lacking because of the difficulty to tune in situ the optomechanical couplings and the direct optical coupling.

The paper presents a quite clear experimental demonstration of enhanced transduction both in the linear and in the nonlinear regime and this is an important results for the field and for science in general because optomechanical sensors can be applied in many scientific and technological areas.

The paper first describes the model and the basic expressions for the power spectral density of the heterodyne detection of mechanical motion, reconsidering older results in the literature, and underlining the fact that the enhancement due to the coupled modes, when the resonance condition are satisfied, is associated with the resolved sideband ratio (mechanical frequency/cavity decay rate).

The key technical advancements allowing to achieve the novel results are: i) exploitation of the material mechanical anisotropy, by tuning the angle θ between the material 010 crystal axis and the optical axis of the device; this allows independent mechanical tuning and optical tuning of the mode properties.

ii) Post-fabrication tuning of the optical mode coupling via a tunable thermal gradient, allowing to control the optical mode splitting and coupling to be adjusted, and to have the desired coupled-mode system (Fig. 2b)

These two technical solutions are novel and relevant, and they represent the basis for the relevant experimental demonstration of enhanced mechanical detection, both in the linear and nonlinear regime, shown in Fig 4 and Fig 5.

I think that the paper deserves publication in Nature Communication because of its relevance and broad interest. Even though the presentation is quite clear and, together with the supplementary information and Methods, provides a quite complete system and methodology description, I think that there are few points that should be fixed before publication in order to improve the results and the presentation.

We thank the reviewer for their careful assessment and the expressed appreciation of our work. We are very glad that the reviewer recognizes the relevance and broad interest of the work. We respond to the specific recommendations below.

1. The model adopted describes very well the linear transduction data (Fig. 4), while it underestimates the effect in the nonlinear case (Fig. 5). This discrepancy is not discussed in a convincing way in the paper and I would provide a deeper discussion of it. The perturbation theoretical treatment adopted to describe the model seems reasonable and appropriate for the given parameters, and in fact, the authors do not attribute the discrepancy to this issue. The fact suggested by the authors that the value of the vacuum optomechanical coupling rates g_L and g_R can be wrongly estimated may be reasonable in my opinion, but I would further investigate this point. The calibration method of Ref. 36 is used for the coupling measurement, but this method assumes the knowledge of the thermal phonon number and this is in my opinion a strong limitation, and a possible reason for the inaccurate estimation of the coupling. In fact, in this case, the thermal gradient tuning adopted may change considerably the local temperature, leading in this way to a wrong estimation of g_L and g_R . The author could profit from the alternative method for the measurement of the vacuum optomechanical coupling rate shown in P. Piergentili et al., Phys. Rev. Applied 15, 034012 (2021) which does not need the knowledge of the mean thermal phonon number. Another possible point to check is the normalization of the calibration with respect to the cavity photon number, which may be sensitive to the value of various parameters measured independently. I have the feeling that the results could be improved and better explained by considering in more details the above aspects, making in particular a critical analysis of the measurement of the optomechanical coupling rates.

We thank the reviewer for bringing up this point, which has inspired important improvement of the manuscript. It motivated us to perform a deeper analysis in the retrieval and error estimation of system parameters such as the optomechanical coupling rates. We performed additional experiments and include the new results, analysis, and discussion in the revised manuscript and the supplementary material. While this did not take away the deviation between model and experiment in Fig. 5 that we already acknowledged in our original submission, it allowed a more accurate experimental analysis of the involved factors and discussion of opportunities for future research. Moreover, we recognize that the model does very well predict the enhancement due to the coupled-cavity configuration, that is the main topic of our work. We could illustrate this in a new figure panel that is added to the revised Fig. 5.

We first address the thermal phonon number that the reviewer mentions as a potential source, which we had also discussed briefly on page 10 of our manuscript. In Supplementary Note 4: ‘Thermal tuning simulation and evaluation’, we use perturbation theory to estimate the sensitivity of the optical resonance to temperature. From the measured optical resonance frequency shift during tuning with the auxiliary laser, we then estimate the temperature increase of approximately 6 Kelvin, making its effect on the thermal phonon number very small, given that our experiments are performed at room temperature. The effect would be an overestimation of 2% of the linear transduction and an overestimation of 4% of nonlinear transduction in the coupled system, with respect to the uncoupled system. This is thus not enough to explain the underestimation of nonlinear transduction by the model.

Next, we have conducted a more elaborate analysis of the possible error on the determination of the vacuum optomechanical coupling rates. In the newly added Supplementary Note 8 'Determining vacuum optomechanical coupling rates', we describe in detail how these coupling rates were determined and estimate how large the uncertainty of our method is given uncertainties on experimental parameters and statistical variations. We performed the experiment for various many different laser detunings, and find a relative standard error on the coupling rates of 3%, which is dominated by the uncertainty in determining the exact V_{π} of our electro-optic phase modulator and the effect of residual amplitude modulation. Note that the effect of thermal phonon number is smaller than these errors: because no tuning is used when measuring vacuum coupling rate, the sample temperature is very close to environment temperature. Again, we conclude that the uncertainty in the determined vacuum coupling rates is not large enough to explain the discrepancy between nonlinear transduction and the model.

The reviewer refers us to the method described in P. Piergentili et al., Phys. Rev. Applied 15, 034012 (2021), which gives an interesting new approach to determining vacuum coupling rate: There, knowledge of the thermal phonon number is not required, as the mechanical resonator is brought into self-oscillation. In that paper, this method is applied experimentally to a free-space optical resonator with a mechanical membrane inside, where it gives a very accurate estimation of vacuum coupling rate. While this is a very interesting approach, we determined that it is unfortunately not easily applicable to our experimental system and conditions: The method requires good knowledge of the optical linewidth and outcoupling rate, which are unfortunately difficult to determine and do not remain constant for the high powers that we require to induce self-oscillations in our photonic crystal system (which has modest mechanical quality factor at room temperature). Such required powers in fact significantly exceed those at which our measurements were taken, and lead to two-photon and free-carrier absorption that would make retrieved parameters not representative. We expect that this could potentially be overcome in the future by moving to a cryogenic environment. In that context, we have added the suggestion and reference to our revised manuscript.

Finally, the referee asks about the normalization of the nonlinear data with respect to the cavity photon number. Here, we note that both the experimental data and the model are scaled by the same expected cavity photon number, which is itself based on the experimentally determined detuning, optical linewidth and optical outcoupling rate. Because of this, the normalization does not change the ratio between the model and the experimental data and thus does not influence the observed deviation. Indeed, this deviation must be related to the ratio between linear and nonlinear transduction, which is controlled only by the vacuum coupling rates, optical linewidths, and thermal phonon occupation. Further potential sources of the deviation that could be considered include the potential existence of thermo-optomechanical forces and backaction that were not considered in our model. We believe that experiments at different temperatures could shed further light on that and/or other mechanisms. We discuss these in the revised Discussion section.

Following the reviewer's recommendation to strengthen the discussion of the results, we also took a closer look at the enhancement of nonlinear transduction that is achieved in the coupled-mode system as compared to the single-cavity system, which constitutes a main finding of our work. Importantly, the model and the experimental data show excellent agreement regarding this enhancement factor. We have added an additional panel (e) to Figure 5, which plots the ratio of optomechanical transduction power of tuned and detuned case for both measured data and our model, as a function of laser

frequency. This shows that the model predicts correctly the change in nonlinear transduction between different laser-cavity detunings and optical mode detunings, and corroborates the reported enhancement by one order of magnitude.

Action taken:

- Added Supplementary Note 8 ‘Determining vacuum optomechanical coupling rates’, which elaborates on the method that was used to determine the coupling rates and analyses the possible uncertainty. This also includes 3 new supplementary figures.
- Added Supplementary Note 9 ‘Tracking of optical linewidth and outcoupling rate during measurements’, which shows and discusses raw data that illustrates how the system’s optical parameters (dissipation and waveguide coupling rates) are tracked and checked during measurements.
- Added new panel e to Fig. 5, showing the relative enhancement due to the coupled-cavity operation as a function of laser detuning, and the model agreement. The caption of Fig. 5 was updated accordingly.
- Text updates in ‘Multimode enhancement...’ section, which are highlighted in the .pdf file that shows the changes to the manuscript during revision.
- Text updates in ‘Discussion’ section, to clarify the estimated 4% effect of temperature increase on nonlinear transduction, the estimation and role of the vacuum optomechanical coupling rates and optical dissipation rates, and discuss in more detail potential causes of experimental deviations from the model predictions, and opportunities for future research. That includes the other calibration method suggested by the referee, in systems where they could be employed.

I have also two final minor points:

(i) In the perspectives the authors mention various quantum applications. I am less optimistic: can the thermal tuning method safely applied in cryogenic scenarios? The mechanical quality factors seem quite low at the moment and the I fear that one needs to improve significantly the quantum cooperativity in order to approach the quantum regime. Nonetheless I agree that operating the present coupled-mode system in the quantum regime would be of extreme relevance.

We agree with the reviewer that cryogenic usage of the thermal tuning method is not clear to work. We address this point in the discussion (page 11), highlighting some of the alternative tuning techniques that are more cryo-compatible or have been demonstrated under cryogenic conditions, including oxidation tuning, light-induced chemical etching or laser-induced gas desorption: “Moving forward, ... laser-induced gas desorption”.

Additionally, we fully agree that the mechanical quality factors in our proof-of-concept room-temperature experiments are relatively low. As we write in the Discussion, they are both limited by radiative and non-radiative decay mechanisms. Cooling the nanobeam down to cryogenic conditions will allow for strong reduction of the non-radiative decay, while the radiative decay introduced by our non-zero fabrication angle can be mitigated using a support structure that has a full phononic bandgap. Indeed, the photonic crystal platform that we chose to work on was one of the first to achieve quantum cooperativity exceeding unity, and remains promising in that regard.

Action taken:

- To clarify the discussion on these topics, we made textual changes to two paragraphs in the Discussion section (end of page 10 and beginning of page 11), emphasizing the benefit of new research on cryo-compatible tuning methods and extending the remarks on how to increase mechanical quality factors.

(ii) In eq (3) and also in the description of the coupled mode model in the Methods section, the notation J is used rather than J_0 for the optical coupling. If I understood properly, this should be only a misprint and not a novel, undefined parameter.

We thank the referee for careful reading of the manuscript and alerting us of these textual errors. Indeed, J is supposed to be J_0 in these cases.

Action taken:

- J in equation 3 of the main text was changed to J_0 .
- In equation 18, 19 and 28 of the Methods section, instances of J were changed to J_0 .

Reviewer #2

Optomechanics studies the radiation pressure interaction between light and mechanical motion. The field has seen rapid advances in the past decade, including both optomechanical devices that enter a variety of quantum regimes and precision optomechanical sensors. A key area of important in the field is the development of new techniques to enhance the light-matter interaction strength, both linear interactions (used, for e.g., for quantum ground state cooling) and nonlinear interactions (proposed, for e.g., to allow the generation of Schrodinger cat and phonon-number states). A long-standing proposal to enhance light-matter interactions is to use a pair of coupled optical cavities. However, because very precisely defined parameters are required for this to work, it has not previously been convincingly achieved. This manuscript convincingly shows the predicted enhancement.

The manuscript is well written and extremely comprehensive, the results are convincing and carefully analyzed. As well as demonstrating a long-predicted and important phenomenon; it introduces significant technical advances, including the development of a new coupled-cavity architecture, a new method of cavity resonance frequency tuning, and the exploitation of the anisotropy of silicon to couple pairs mechanical modes. I have no hesitation in recommending publication in Nature Communications.

We thank the reviewer for these nice words on the manuscript, the impact of the main results, and the technical advances. We are happy that the reviewer recommends publication in Nature Communications.

I have only a few comments:

1) In paragraph 2 (starting on line 32), the authors claim that the regime of $g_0 > \kappa$ has not yet been reached. This is strictly, not true. It has been reached in atom-optomechanical systems (e.g. Science, 322 235–238 (2008)).

We thank the referee for this remark and fully agree that it is important to mention such work in our introduction. We do want to highlight that another crucial ingredient for many applications is sideband resolution, a condition that is not met in the atom-optomechanical systems.

Action taken:

- Changed end of second paragraph to explicitly mention the atom-optomechanical systems reaching $g_0 > \kappa$, including citation.

2) Some of the conclusions of the paper based on Fig. 3 seem unsupported by data due to the inaccessible hatched parameter regions on the figure (where data taking wasn't possible due to optomechanical instability). E.g. on line 5 of pg 6 it's claimed that the signal peaks correspond to the resonant conditions for the upper sideband, laser, and lower sideband. However, the laser-resonant peak appears not to be included in the data - this is within the hatched region if I'm not mistaken. Similarly, for line 29, it appears that the peaks for $\pm \mu J_0$ may both be in the hatched region. It's important that the authors address this. I would suggest, also, including hashed vertical lines on the experimental PSDs in Fig. 3 to show the key resonance conditions.

The reviewer is right in noting that the laser-resonance condition is not fully visible in the data of Figs. 3a and b, because this specific laser-cavity detuning cannot be reached in the direct-detection setup because of a thermo-optic bistability. In the presence of this bistability, however, it is however still possible to closely approach the laser resonance from the blue side, i.e. with laser frequency decreasing towards cavity resonance. This does allow to draw conclusions about optomechanical transduction close to resonance. In this way, the condition $\Delta = 0$ is at the rightmost edge of the hatched region, while in Fig. 3b, the conditions $\Delta = \pm J_0$ are the right edge of the hatched regions. We realize that our precise wording was confusing and could be significantly improved such that we no longer refer to 'at laser resonance', but 'approaching laser resonance' in the revised text. We have now also added the vertical dashed lines to indicate these conditions in Figs. 3 a-c.

We would like to note that the heterodyne detection method used in Figs. 4 and 5 allows much smaller optical power, which avoids the thermo-optic bistability. It is therefore possible to achieve the resonance condition $\Delta = J_0$ in this setup, and the purple curve in Fig. 4a displays the measured signal at this specific resonance condition. In this way, it complements the data taken in Fig. 3.

Action taken:

- Vertical hashed lines were added to Figure 3 a, b and c to indicate the key resonance conditions, and adapted figure 3 caption.
- Changed wording in main text to no longer make claims about exact laser-cavity resonance.

3) In the paragraph on page 8, line 81, the authors suggest that a systematic error might explain the deviation of their nonlinear transduction data from their model. However, I would have thought that such an error this would also degrade the (very good) agreement for linear transduction?

The reviewer is right that an arbitrary error on the parameters may also affect the predicted linear transduction profile, although it is important to remember that the linear transduction was used to fix the global scaling between model and data by matching both at the point of largest transduction. Importantly however, we recognize that the deviation we observe relates to the *relative* amplitude of

nonlinear to linear transduction, which is governed by the vacuum optomechanical coupling rates, the optical linewidths, and the thermal occupancy. Thus, what we referred to in the original text is that a systematic underestimation of the vacuum coupling rates or thermomechanical motion, or overestimation of the optical linewidth, could result in an overall underestimation of nonlinear transduction in accordance with the observed deviation. That being said, inspired by the related first question of reviewer #1, we made several adaptations to the manuscript to further investigate and discuss the observed deviation. Moreover, we note that the discussed deviation concerns the absolute amplitude of the nonlinear transduction. Its enhancement through tuned coupled-cavity interactions is in fact corroborated quantitatively by the model, as we illustrate in a new figure.

Action taken:

- See for full revisions the response to question 1, reviewer 1. We made several adjustments to the revised text including in particular a detailed analysis of the parameter estimation and its impact in the main text and two new Supplementary Notes, and added Fig. 5e to quantify the relative coupled-mode enhancement of optomechanical nonlinearity in model and experiment.

4) The authors show enhancement of both the linear and nonlinear optomechanical interactions. However, for the nonlinear optomechanical interaction to be effective in preparing non-classical states such as phonon-number states, it is generally crucial that the linear interaction is very strongly suppressed - the linear interaction introduces decoherence and typically has a much higher magnitude than the nonlinear interaction. It would be helpful if the authors could comment in the discussion about the prospects of non-classical state-generation in the presence of the linear coupling. Would it be necessary to suppress this? If so, by how much, and how can it be done? If not, why not.

Indeed, the residual linear coupling is the most important aspect in many applications of nonlinear optomechanical interactions. In a previous work (Burgwal et al., New Journal of Physics 22 113006 (2020)), we theoretically investigate the coupled mode system and conclude that the nonlinear enhancement in coupled-cavity systems is accompanied by linear transduction in the weak coupling regime. Interestingly, though, when only one optical supermode is populated, the linear and nonlinear sideband occupy orthogonal optical modes, which means they can be easily separated. This can be useful in classical applications, although it does not help in mitigating quantum backaction noise due to linear coupling.

In our discussion, we make a distinction between the weak coupling regime, where linear coupling is significantly larger, and the strong coupling regime, where the two types of coupling are of comparable size. In the weak coupling regime, nonclassicality can be created even in the presence of residual linear coupling by using heralding, which uses a single photon detector as the strongly nonlinear resource, whereby the coupled-cavity configuration enhanced the heralding rate of multi-phonon states (see Burgwal et al., New J. Phys. 22 113006 (2020)). In another approach, Brawley et al., Nat. Commun. 7 10988 (2016), discuss how decoherence due to residual linear transduction can be suppressed by employing a feedback mechanism.

For the strong coupling regime, we refer to several works which have taken into account the coupled cavity linear transduction (i.e. Ludwig et al., PRL 109 063601 (2012) and Stannigel et al., PRL 109, 013603 (2012)) in their analysis of the generation of nonclassical optical or mechanical states, confirming that

this is possible as long as the strong coupling regime is reached and that the coupled mode system can help increase the degree of nonclassicality.

Action taken:

- Rewritten final paragraph of discussion and added a concise discussion of residual linear coupling and the need to mitigate this for nonclassical state generation applications.

OTHER CHANGES TO MANUSCRIPT:

- Improved fitting routine of nonlinear transduction data, data for uncoupled cavities now more reliable.
- Made various typographical corrections throughout, and made the ordering of symbols in products in equations more consistent.

REVIEWERS' COMMENTS

Reviewer #1 (Remarks to the Author):

The second version of the paper is significantly more detailed than the first one, and the authors have satisfactorily addressed all the comments I had in my first review. The effect of thermal tuning has been satisfactorily discussed, and also the question of the measurement of the coupling rates is now discussed in detail. I think that now the results are convincingly described, and I suggest publication of the paper in the present form.

Reviewer #2 (Remarks to the Author):

The authors have addressed both reviewers comments to my satisfaction. I recommend publication as is.